# Hops Germplasm: Phytochemical Characterization of Wild *Humulus lupulus* of Central and Northern Italy

**DOI:** 10.3390/plants11121564

**Published:** 2022-06-14

**Authors:** Lisetta Ghiselli, Lorenzo Marini, Cosimo Taiti, Luca Calamai, Donatella Paffetti, Elisa Masi

**Affiliations:** Department of Agriculture, Food, Environment and Forestry (DAGRI), University of Florence, Piazzale delle Cascine 18, 50144 Florence, Italy; cosimo.taiti@unifi.it (C.T.); luca.calamai@unifi.it (L.C.); donatella.paffetti@unifi.it (D.P.); elisa.masi@unifi.it (E.M.)

**Keywords:** *Humulus lupulus*, wild hop, hop germplasm, bitter acids, polyphenols

## Abstract

Hops are widespread as a wild plant in almost all Northern and Central Italy, and the characterization of wild populations is attracting considerable interest in verifying their potential use. The development of hops as agricultural crop can be an interesting opportunity, both for farms that would have available a new crop to be included in the crop system and for craft breweries interested in characterizing beers with local raw materials. In the present work, 14 wild hop accessions coming from various Italian locations were characterized and compared with 2 commercial varieties (Cascade and Hallertau Taurus) grown in the same environments. The cones were analyzed to measure the content of α- and β-acids, polyphenols, flavonoids, and the anti-radical power. The α-acid content of wild hops was generally low, while the β-acid content was very variable and quite high in some samples. The content in polyphenols and flavonoids and the antiradical power were high and generally similar to those of the commercial varieties. Therefore, the analyzed genotypes are not very suitable for use as bitter hops in beer production, while further analysis may indicate a possible use as aroma hops, or for herbal and pharmaceutical purposes, thanks to their antioxidant content.

## 1. Introduction

Hops (*Humulus lupulus* L.) is a rhizomatous, climbing, dioecious herbaceous plant belonging to the family of the Cannabaceae and originating from a vast area including Europe, Western Asia, and North America. In our environments, the plant develops new shoots at the beginning of spring, fructifies during the summer, while in autumn the vegetative part withers due to the cold, leaving the rhizome in the soil, thus conferring a perennial character to the plant. The production of the hop plant consists of female inflorescences called cones, which together with cereals, water, and yeast are the raw materials used in the production of beer. The importance of hops in brewing is derived not so much from the amount of cones used, which is rather limited, as from their quality. In fact, they contain several substances that can impart to beer well-defined aromas and a bitter taste, as well as increase the shelf life and the stability of the foam, and promote protein coagulation. Therefore, the hops, more than the other ingredients, can contribute to the characterization of beer and to create a link between product and environment of origin of raw materials. In this sense, even for beer it becomes possible to talk about terroir as it happens for wine. The type of hops, understood both as genotype and as a cultivation environment (geographic area and climate), allows to obtain a specific and identifiable beer through the unique characteristics of its territoriality. Today, at a national and, even more, at a regional level, there is a need for craft brewers to make use of local raw materials to characterize the beer product. This is in line with what is happening in some Northern European countries where beer has always been a traditional and widely consumed beverage, and where there has long been a tendency to use local ingredients.

Among the substances present in the cones that arouse the greatest interest in brewing are definitely the amaricant compounds, the essential oils, and the polyphenols, whose percentages are variable according to the genotype, the growing environment, the harvesting period, and also the temperatures of storage [1,2,3,4,5,6]. The characteristic data of a hop with regard to bitterness is its percentage of α-acids, while β-acids percentage are essentially linked to taste and aroma.

Alpha acids include three main molecules: humulone (35–70% of total α-acids), cohumulone (20–65%), and adhumulone (10–15%), which are also known for their anti-inflammatory and antioxidant properties [7,8]. When the part of cohumulone on the total of the α-acids is less than 30%, the cultivar of hop is basically defined aromatic; if instead it is more than 30%, the cultivar is defined as bitter. Beta-acids include instead: lupulone (30–55% of total β-acids), colupulone (20–55% of total β-acids), and adlupulone (10–15% of total β-acids); they are very sensitive to oxidation and degrade during the brewing process and are also almost completely insoluble in water [7,8]. 

Regarding the polyphenolic component, in recent years, studies have shown the effectiveness of using hop extracts against some chronic-degenerative diseases, such as cancer and metabolic syndrome [9]. This perspective has prompted some researchers to investigate the use of agricultural waste from the hop harvest as a source of secondary metabolites for use in industries other than brewing, such as food, cosmetics, and pharmaceuticals [10]. Hop is a very widespread species in the wild state in Central and Northern Italy, but its cultivation is not strongly developed. In Italy, the hop cultivation areas in 2020 was limited to 52 hectares, distributed mainly in the Northern regions [11].

From the characterization of national wild germplasm, some genotypes have been classified as dual-purpose hops or flavoring hops [12], which among other things are the most sought-after hops on the market. The development of hops as agricultural crop can be an interesting opportunity, both for farms that would have available a new crop to be included in the crop system and for craft breweries interested in characterizing beers with local raw materials. With this in mind, there is an increasing interest in the development of cultivars or varieties suitable for Italian environments, starting also from ecotypes of wild germplasm identified by aromatic and production characteristics. Therefore, the objective of this research is to obtain a first classification of the Italian wild hop germplasm for some phytochemical compounds compared to commercial hops cultivars. This initial characterization allowed us to identify genotypes suitable for brewing and to increase the quantity by their cultivation.

## 2. Results and Discussion

It is known that wild hops constitute a potential genetic resource from which to obtain new germplasm to increase the variability useful for genetic improvement. Northern-Central Italy is rich in wild hops, although this species has not been able to take off as a crop and become fully part of the crop plans of Italian farms. Therefore, in this work an initial characterization of 14 wild hop genotypes, collected in Central and Northern Italy, was carried out and compared with 2 commercial hop varieties. All genotypes were grown in two areas of Central Italy, and were evaluated for some phytochemical compound (polyphenols, flavonoids, ARPs, α-acids, and β-acids) in hop cones.

### 2.1. Polyphenols, Flavonoids and ARP

Secondary metabolites content, as well as antioxidant capacity (ARP), of the various hop genotypes, both determined using a colorimetric method, are presented in Table 1. Of these parameters, only free and total flavonoids were significantly different among the different genotypes and showed a similar trend. With regard to the free fraction of flavonoids, the content varied from about 6.07 and 6.22 mg/g DW of the Se and H.T/T samples to 14.79 mg/g DW of the T/S sample (significant means for *p* < 0.05). The total flavonoids content ranged from 11.77 (H.T/T sample) to 24.23 (S/S sample) and 26.14 (T/S sample) mg/g DW with statistically significant differences (*p* < 0.05) between samples. 

As previously noted [13], the free fraction of polyphenols, of the antiradical power and, although to a lesser extent, of the flavonoids, are more representative than the bound fraction. Moreover, from the evaluation carried out, the mean values of polyphenols, flavonoids, and ARP of commercial cultivars were very close to that found in wild germplasm, contrary to what was found in a previous study [13], where the differences in favor of wild germplasm were more pronounced. These differences are probably due to the environmental differences: those where the wild germplasm was collected and those where genotypes were grown. It should also be noted that the values obtained for the various secondary metabolites and the anti-radical power both for the cultivars and for the wild germplasm provided mean values lower than previously noted [13]. This is probably due both to climatic and environmental conditions and, above all, to different genotypes. 

### 2.2. Bitter Acids

By means of HPLC analysis, quantitative and qualitative analyses of α- and β-acids were carried out in the resins of the hop cones. In particular, the compounds detected were: cohumulone, humulone + adhumulone, colupulone, lupulone + adlupulone, and total α- and β-acids. The percentage of cohumulone on total α-acids, the percentage of colupulone on total β-acids, and the α/β acid ratio were also calculated.

The α-acids, responsible for the bitterness, were on average significantly higher in commercial varieties (9.15 g/100 g DW) compared to the wild germplasm (2.03 g/100 g DW) (Table 2). Among other things, the content of α-acids of wild genotypes was on average higher than that found (1.72 g/100 g DW) on 23 Italian wild genotypes [14], while it was lower than that reported in other assessments of both national germplasm (3.82 g/100 g DW and 3.07 g/100 g DW) [12,15] and European germplasm (2.38 g/100 g DW) [16], respectively. These differences, as already indicated above for secondary metabolites, highlight the high variability of the wild germplasm. In the commercial varieties, the contents were on average variable from 6.75 g/100 g DW of the Cas/T sample (Cascade grown in Tagliacozzo) to 9.56 g/100 g DW of the Cas/S sample (Cascade grown in Subiaco) and 10.94 of the H.T/T sample (Hallertau Taurus grown in Tagliacozzo). The Cascade variety had a mean α-acids content of 8.15 g/100 g DW and was in line with commercial values, while the Hallertau Taurus variety had a value of α-acids (10.94 g/100 g DW) which was below the commercial ranges reported for this variety (12–17%).

In wild genotypes, the values of α-acids were variable from 0.12 to 3.16 g/100 g DW, respectively, of the Sus/T (cultivated in Tagliacozzo) and T/S (sample cultivated in Subiaco) samples. On average, the wild hops cultivated in Subiaco (average data 3.42 g/100 g DW) had higher α-acid contents than those, also wild, cultivated in Tagliacozzo (average data 2.78 g/100 g DW).

With regard to the single compounds of α-acids, the cohumulone levels were higher in the cultivated varieties with an average content of 2.68 g/100 g DW, compared to the ecotypes whose average content was 0.13 g/100 g DW (Table 2). In particular, the Cas/S cultivar showed the highest content of cohumulone and differs statistically significantly from the others. Moreover, for the content of humulone + adhumulone (Table 2), the commercial varieties were those with the highest quantities, and on average their contents (6.47 g/100 g DW) were three times higher than the average of the ecotypes (1.90 g/100 g DW). Among the commercial varieties, the content was high in particular in H.T./T (8.58 g/100 g DW); for wild germplasm, the highest average values were those found for T/S ecotype (3.02 g/100 g DW), while the lowest values were detected in the wild genotypes cultivated in Taglacozzo (mean of 1.30 g/100 g DW).

Total β-acids (Table 2) were also found to be on average higher in commercial varieties (5.57 g/100 g DW) compared to ecotypes (2.48 g/100 g DW). Among the latter, it is necessary to highlight the Sus/T sample that showed the greatest contents of β-acids (4.01 g/100 g DW), and statistically not different from Cas/T and H.T/T commercial varieties. The β-acid content in the analyzed wild germplasm was on average higher than the values reported [14] for Italian germplasm samples (2.48 versus 1.57 g/100 g DW). The values are quite close to those reported by another selection of Italian germplasm [12,15], and lower than those of European germplasm (2.48 versus 2.59 and 3.66 g/100 g DW, respectively) [16].

The level of colupulone (Table 2) was on average higher in commercial varieties (2.91 g/100 g DW) compared to wild genotypes (0.93 g/100 g DW). Similarly, the average content of lupulone + adlupulone (Table 2) was higher in commercial varieties (2.67 g/100 g DW) than the average wild germplasm (1.54 g/100 g DW). Among these, the three wild genotypes S/T, and Sus/T above mentioned showed values of lupulone + adlupulone (2.27 and 2.67 g/100 g DW, respectively) higher or equivalent to levels found in commercial varieties, where the levels ranged from a minimum of 2.24 for H.T./T to a maximum of 2.91 for Cas/S.

Another very important parameter used to evaluate commercial hops is the percentage of cohumulone on the total content of α-acids (Table 2). In the commercial varieties, this parameter was found to vary from a minimum of 21.4% of H.T/T to a maximum of 34.4% of Cas/S; the values are in line with those reported in the literature for other cultivated varieties [12]. Wild genotypes showed very variable values, from a minimum of 0.5% of the Se sample to a maximum of 19.2 and 27.3% of the S/T and the Sus/T samples, respectively; these latter values are close to those reported for wild hops [16] and commercial varieties [12]. The 30% threshold for cohumulone, out of the total alpha acids, marks the limit between aroma hops and bitter hops; from the data obtained, it is evident that wild genotypes are predominantly aromatic hops. However, the average 8.6% for wild germplasm was considerably lower than the average (23.3%) obtained in other European wild hops [16].

The percent content of the colupulone on the total β-acids is another parameter taken into account for the evaluation of commercial hops [12]. This parameter was variable for the varieties analyzed from a minimum of 47.2% of H.T./T to a maximum of 54.8% of Cas/S; the values are in line with what is reported in the bibliography from other studies [12]. For this parameter, the wild germplasm evaluated in our study provided values variable from 33.5% of the Sus/T and Che/T samples to 44.9% of the Te sample (Table 2). The average figure of 38.2% was in line and slightly lower than that obtained by other researchers for the European wild germplasm: 41.9% [16] and 41.1%, respectively [12].

The α/β acid ratio is considered a very important index for the definition of the quality of bitter hops [12]. From the observations made on the samples analyzed, it was possible to detect, for the cultivated varieties, a high mean value of the α/β ratio ranging from 1.2 of Cas/T to 2.6 of H.T./T (Table 2). In wild germplasm, the α/β ratio varied from a minimum of 0.03 of Sus/T to a maximum of 1.6 of Se sample and these values are close to those reported in the literature for wild Italian hops [15]. The highest contents for bitter acids in wild germplasm were found in the T/S sample for α-acids (3.2 g/100 g DW and α/β ratio 1.2) and in the Sus/T sample for β-acids (4.0 g/100 g DW and α/β ratio 0.03). The T/S sample provided, among the analyzed wild genotypes, the highest total acid contents, equal to 5.6 g/100 g DW.

In order to have a graphical representation of the distribution of wild and cultivated hops based on phytochemical profiles, all quantitative variables measured were subjected to PCA analysis using R-based software. The first two major components (PCs) used in this study almost explain 87% of the total variance (Figure 1). The graphical representation shows the distribution of the samples along the two PCs and the contribution of the individual variables on the two PCs. In particular, it is possible to distinguish, in the two-dimensional distribution of the samples analyzed, the group of commercial hop varieties that tend to be placed in the right of the plot, as characterized mostly by high values of α and β acid as single components and also as α/β ratio. These are all parameters that together with total flavonoids contributed most to determine the first PC. The H.T./T commercial variety tends to deviate from the Cas group (Cas/S and Cas/T) especially for the higher α/β acid ratio (2.6 versus 1.3). Instead, the percentage of lupulone and cohumulone in total α acids variables was the one that weighed most in the definition of PC2.

Wild germplasm samples instead go to the left side of the plot and among them we can distinguish three groups: (1) includes Sus/T and S/T samples characterized by higher lupulone (2.0 and 1.6 g/100 g DW, respectively) and cohumulone percentage in total α acids (27.3 and 19.2%, respectively) values compared to the average ones (1.2 g/100 g DW and 8.6%) found in wild germplasm; (2) includes T/S, Che/T, S/S, T/T samples, and in part also Te sample (this sample seems to be halfway between the second and third group) characterized for the humulone + adhumulone, lupulone + adlupulone, lupulone, and colupulone percentage in total β acids contents; and (3) includes Te and Se samples characterized by higher α/β ratio (1.4 e 1.6 g/100 g DW) values compared to the average content of wild germplasm (1.0 g/100 g DW).

## 3. Materials and Methods

### 3.1. Plant Material

In the experimentation, cones collected in 2016 from hop plants of two germplasm collections in the following localities were analyzed: municipality of Tagliacozzo (province of L’Aquila, 42°4′10″ N, 13°15′17″ E) at 740 m a.s.l. and municipality of Subiaco (province of Rome, 41°56′ N, 13°06′ E) at 408 m a.s.l. Wild material was collected according to the characteristics of the cones similar to the cultivated type. The germplasm collections included both wild plants of Central-Northern Italy (14 genotypes) and plants of 2 commercial varieties. The genotypes analyzed were not replicated homogeneously in the two locations of cultivation, because they belonged to private collections. Therefore, for comparison purposes, homogeneous groups were obtained by combining the origin of the genotype with the cultivation locality (Table 3). 

Cones were harvested for each genotype, for both locations, at two successive times (15 days apart) in September 2016. These two samples represented the experimental repetitions analyzed. These samples were immediately dried in a ventilated oven at a temperature of 50 °C for 24 h in order to bring the moisture content around 10%, and then stored under vacuum. Samples were subdivided into two parts: one part was lyophilized and subsequently ground to analyze the content of the free, bound, and total fraction of polyphenols, flavonoids, and antiradical power; and the other part was reduced to powder by mortar with the addition of liquid nitrogen and intended for analysis of α- and β-acids.

### 3.2. Chemical Reagents

Methanol (HPLC grade), formic acid, absolute ethanol, and Folin–Denis reagent were used. Ultra-pure distilled water was self-produced by a Milli-Q device (Millipore, Bedford, MA, USA). It has also been used a bitter acids mixture standard (International Calibrations Extract ICE-3) specific for the quantitative analysis of α- and β-acids in hops and distributed in Europe by the company Labor Veritas Co. (Zürich, Switzerland). All other chemicals and solvents used were of analytical quality.

### 3.3. Analysis of Polyphenols, Flavonoids and Antiradical Power

Phenolic compounds were extracted from each sample of hops following the method of [17] which provides for a first extraction of the free fraction in 80% ethanol, followed by extraction of the bound fraction using first an acid solvent and then an alkaline one.

Polyphenols, free and bound, were analyzed using the Folin–Denis method using a UV/VIS spectrophotometer (Lambda 25 spectrophotometer, Perkin Elmer Corporation, Waltham, MA, USA), using gallic acid as a reference standard [17]. The content of the flavonoids, free and bound, was determined in an analogous way using the colorimetric method with catechins [18]. The total content of polyphenols and flavonoids was obtained for both by the sum of the free and bound fraction. The antioxidant activity was evaluated on the basis of the antiradical activity of the bioactive compounds present in the free and bound extracts, using the stable radical 2,2-diphenyl-1-picrylhydrazyl (DPPH) according to the spectrophotometric method of Brand-Williams et al. [19]. The 2,2-diphenyl-1-picrylhydrazyl was reduced to 2,2-diphenyl-1-picrylhydrazine (DPPHH). The antiradical activity of the samples was first expressed in effective concentration (EC50), then in anti-radical power (ARP). Specifically, EC50 represents the amount of antioxidant present in a sample necessary to decrease the initial DPPH concentration by 50% (EC50 = (mol/L) AO/(mol/L) DPPH). The EC50 is a parameter widely used to express the antioxidant power of a matrix; in this study it was expressed, at first, as mg/g s.s. of extract in relation to the initial concentration of DPPH. The EC50 of each extract was measured for each sample by comparing it to the calibration line by means of linear regression, using solutions with different DPPH concentrations. The higher the EC50 value, the lower the antioxidant activity of the sample. The results of each sample, for greater clarity, have been converted into anti-radical power (ARP), a parameter that directly expresses the antioxidant power (a higher value of ARP corresponds to a greater antioxidant power of the sample), according to the equation: ARP = (1/EC50) × 100.

### 3.4. Analysis of Alpha and Beta Acids

From each sample of appropriately ground cones, an aliquot of 0.040 g was weighed within 2 mL microtubes and subjected to extraction with 1.5 mL of organic solvent consisting in an aqueous mixture of 80% of methanol (CH3OH), and 0.5% of formic acid (HCOOH). Subsequently the microtubes were vortex mixed and incubated in an ultrasonic bath sonicator (Bandelin Sonorex Super RK 102H) for 15 min for three times. Microtubes were incubated at 30 °C with a rotary shaker at 10 revolutions per minute for 24 h. The sample thus extracted was centrifuged at 3000 rpm for 15 min. Subsequently, the supernatant was analyzed with HPLC using the Varian POLARIS C-18A 15 × 2 mm column at a wavelength λ of 326 nm [12]. In order to highlight and separate in the chromatogram all the peaks of the amaricants: cohumulone, humulone, and adhumulone for the α-acids and colupolone, lupulone, and adlupulone for the β-acids, the concentrations of the solvents were modulated over time.

### 3.5. Statistical Analysis

The data obtained from the chemical analyses were subjected to analysis of variance (ANOVA) to estimate the difference between genotypes. The statistical software used was IBM SPSS Statistics version 26. In order to have an overview of the quantitative results for the secondary metabolites and the data resulting from the analysis with HPLC, all 47 samples of hops were subjected to multivariate statistical analysis by means of the analysis of the main components (PCA) using the program R Studio [20], and particularly the packages: factoMineR [21]; factoextra [22]; corrplot [23]; tidyverse [24]. PCA analysis considered only those parameters that turned out statistically significant in ANOVA analysis.

## 4. Conclusions

The present research has evaluated the phytochemical characteristics of wild hops from Central and Northern Italy in comparison with some commercial variety of hops used in the production of beer and well known for their bitter properties. 

From the results obtained, it is evident that the phytochemical profile is influenced by the genotype. Italian genotypes of wild hops were characterized by low values of α-acids and therefore classified as non-bitter hops; in addition, the association of similar levels of α- and β-acids is not always verified, in fact some genotypes such as Sus/T and S/T, although characterized by low content of α-acids, have a β-acid content comparable to that of commercial varieties (Cascade and Hallertau Taurus classified as dual-purpose hops). This result shows the existence of wild hop genotypes suitable for brewing. The variability of phytochemical compounds found in wild hops germplasm and the quantity of some components confirm the importance of this wild germplasm and the possibility to identify interesting ecotypes to introduce in genetic breeding programs and the selection of cultivars suitable for Italian environment. Therefore, the possibility to develop suitable hop genotypes in our country has interesting implications, either regarding craft beer production, now an expanding sector, or in the pharmaceutical aspect, due to the known therapeutic properties of hops. Accordingly, further investigation on the characterization of polyphenols and of the aromatic profile of wild cones, as well as of the beers obtained using such samples, would be precious. This would help in unraveling the relation between raw materials and the sensorial properties of beer.

## Figures and Tables

**Figure 1 plants-11-01564-f001:**
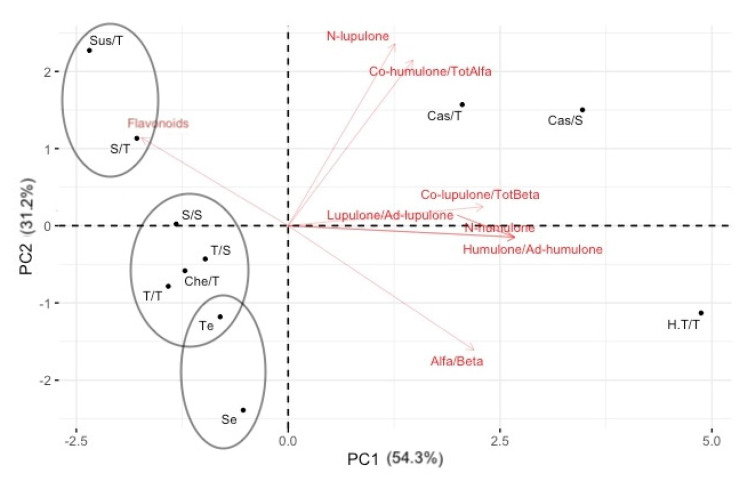
Biplot of the Principal Component Analysis applied to quantitative data of all hop samples for the first two principal components.

**Table 1 plants-11-01564-t001:** Polyphenol and flavonoid contents, and antiradical power (ARP) of all hop samples (cones) considered in this study.

Genotypes	Polyphenols			Flavonoids			ARP		
	Free	Bound	Total	Free	Bound	Total	Free	Bound	Total
	mg/g DW			mg/g DW					
Cas/S	26.93	15.39	42.32	9.77 ab	7.31	17.08 ac	205.97	183.16	389.12
Cas/T	28.15	24.71	50.71	8.92 ab	9.85	18.63 ac	208.57	182.99	391.57
Che/T	22.87	14.36	37.17	10.39 ab	8.47	18.89 ac	207.38	179.96	387.37
H.T/T	24.25	20.79	40.17	6.22 b	5.55	11.77 c	205.97	172.20	378.12
S/S	26.89	17.67	44.18	13.94 ab	10.35	24.232 a	214.84	188.68	403.55
S/T	23.44	18.59	42.02	11.45 ab	10.59	22.04 ab	209.05	182.67	391.71
Se	20.39	14.84	35.23	6.06 b	6.59	12.65 bc	199.92	174.00	373.91
Sus/T	19.18	16.85	35.61	11.57 ab	9.77	21.49 ac	212.13	181.09	393.17
T/S	29.31	14.69	45.17	14.79 a	11.17	26.14 a	227.44	199.64	427.08
T/T	22.30	15.47	38.06	9.85 ab	9.91	19.82 ac	208.87	176.95	385.83
Te	22.46	13.92	36.82	8.28 ab	10.25	18.46 ac	201.40	166.87	368.27
	n.s.	n.s.	n.s.	*	n.s.	*	n.s.	n.s	n.s.

Means followed by a different letter indicate significant differences at *p* < 0.05.

**Table 2 plants-11-01564-t002:** Average bitter acid content in dry cones hops of wild Italian germplasm and commercial cultivar, determined by HPLC.

Genotipes	Total α Acids	Cohumulone	Humulone + Adhumulone	Total β Acids	Colupulone	Lupulone + Adlupulone	Coh % in Total α Acids	Col % in Total β Acids	α/β Ratio
g/100 g DW									
Cas/S	9.56 A	3.28 A	6.28 B	6.43 A	3.52 A	2.91 A	34.38 A	54.85 A	1.49 B
Cas/T	6.75 B	2.11 B	4.64 C	5.60 AB	2.88 A	2.73 AB	31.39 A	51.58 AB	1.21 BC
Che/T	2.32 CD	0.04 C	2.28 DE	2.37 CE	0.78 CD	1.59 BD	6.61 CE	33.38 E	0.94 BC
H.T/T	10.94 A	2.36 B	8.58 A	4.256 BC	2.02 B	2.24 AC	21.42 AC	47.22 AC	2.62 A
S/S	2.32 CD	0.05 C	2.26 DE	2.98 CE	1.11 CD	1.87 AD	5.27 CE	36.67 DE	0.99 BC
S/T	1.29 DE	0.30 C	0.99 EF	3.48 CD	1.21 CD	2.27 AC	19.17 AD	35.76 DE	0.56 CD
Se	1.96 CD	0.01 C	1.95 DE	1.25 E	0.52 D	0.73 D	0.52 E	4.14 CE	1.62 B
Sus/T	0.12 E	0.03 C	0.09 F	4.01 BC	1.35 BC	2.67 AB	27.28 AB	33.64 E	0.03 D
T/S	3.16 C	0.14 C	3.02 D	2.49 CE	0.97 CD	1.52 BD	3.53 DE	37.85 DE	1.23 BC
T/T	1.70 CE	0.16 C	1.54 DF	1.91 DE	0.80 CD	1.11 CD	7.79 CE	39.88 CE	1.00 BC
Te	1.84 CD	0.26 C	1.58 DF	1.46 E	0.66 CD	0.80 D	11.56 BE	44.89 BD	1.41 B

In column different capital letters indicate statistically significant differences mean at *p* ≤ 0.01 by Duncan test.

**Table 3 plants-11-01564-t003:** Hops germplasm considered in this study.

Code	Location of Origin/Cultivation	Commercial Varieties	Wild Germplasm	Number of Samples
**Se**	Subiaco (RM) mother plant		x	4
**Te**	Tagliacozzo (AQ) mother plant		x	5
**S/S**	Subiaco/Subiaco		x	10
**S/T**	Subiaco/Tagliacozzo		x	5
**T/T**	Tagliacozzo/Tagliacozzo		x	6
**T/S**	Tagliacozzo/Subiaco		x	5
**Che/T**	Cherasco/Tagliacozzo		x	3
**Sus/T**	Susa/Tagliacozzo		x	2
**Cas/S**		Cascade		3
**Cas/T**		Cascade		2
**H.T./T**		Hallertau Taurus		2

S = Subiaco (Rome), T = Tagliacozzo (L’Aquila), Che = Cherasco (Cuneo), Sus = Susa (Turin).

## Data Availability

Not applicable.

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
