# Peer review of "Hops Germplasm: Phytochemical Characterization of Wild Humulus lupulus of Central and Northern Italy"

_plants, 2022, doi:10.3390/plants11121564_

Round 1

Reviewer 1 Report

The article does not clearly explain what is meaning "spontaneous". What is the difference between "spontaneous" and "wild" hop plants?

Analysis of hop essential oils should be performed to characterize hop cultivars, especially in relation to the sensory quality of beers. There is no mention of these in the paper.

Analyses of hop polyphenols can also be performed by group and specific (LC-MS/MS) methods. Utilisation only group methods is unsufficient.

Author Response

Please, see attached file

Reviewer 2 Report

Please, see the attached file

Author Response

Please, see the attached file
